# Revisiting Conditional Whitney Forms: From Structure Preservation to Physics Recovery

## Abstract

Conditional Whitney forms have recently emerged as a promising framework at the intersection of scientific machine learning and finite element analysis. They offer a solid theoretical foundation for enforcing conservation laws in complex machine learning settings. However, their use so far has been restricted to learning tasks where structural constraints can be satisfied with simple, yet inaccurate, physics representations. In this work, we analyze why existing formulations reduce to typical unconstrained reformulations, circumventing physics recovery, and highlight the necessity of incorporating additive structure pertaining to the governing physics of the system. Based on the theoretical insights we first attain, we proceed to the reformulation of the learning problem to enable data-driven physics recovery and employ conditional Whitney forms to turn a Transformer-based architecture into a structure-preserving reduced-order model. We demonstrate the validity of our theoretical insights and the effectiveness of the subsequent proposed reformulation in a range of advection-diffusion systems of increasing difficulty. Our contributions can be viewed as a step towards understanding the capacity of conditional Whitney forms to build reliable structure-preserving models by harnessing the modeling power of state-of-the-art machine learning architectures in physical sciences.

## 1 Introduction

Partial differential equations (PDEs) form the foundation for describing and modeling complex systems across engineering and physical sciences. Traditional PDE solvers, such as the finite element method (Brenner and Carstensen, 2004), are computationally intensive, lack generalizability across different PDE parameters, and require customized solver configurations for each specific problem. Moreover, they struggle in settings where the governing physics is only partially known or where only general physical principles apply. Motivated by the achievements of machine learning in fields such as natural language processing (Koroteev, 2021; Radford et al., 2019) and computer vision (Deng et al., 2009; Goodfellow et al., 2014; Ren et al., 2016), operator learning (Kovachki et al., 2024) emerged as a powerful learning paradigm for learning maps between infinite-dimensional function spaces. As neural operators overcome the limitations of traditional solvers and show impressive predictive capabilities, they have been largely employed as surrogates in PDE solving.

Initially, operator learning architectures focused on the construction of appropriate PDE solution spaces, while recent advancements highlight a paradigm shift toward modern deep learning, including Transformer-based architectural designs (Dosovitskiy et al., 2020; Jaegle et al., 2021), continuous neural representations (Stanley, 2007; Mildenhall et al., 2020) and transfer learning via foundation models (Touvron et al., 2023; Radford et al., 2021). Leveraging attention mechanisms (Vaswani et al., 2017), Transformer models can capture multiscale relationships within data representations, while neural fields provide an efficient framework for encoding continuous fields at arbitrary resolutions, aligning closely with the central goals of operator learning. In addition, inspired by the abundance of available data and the recent successes of foundation models, the first scientific foundation models (Bodnar et al., 2025) have been developed. Pre-trained in a vast corpus of diverse physics tasks, these models provide enhanced generalization in downstream tasks with only a modest amount of fine-tuning.

While more and more emphasis is placed on the predictive performance and the efficiency of neural operators, important aspects of the traditional theory of finite element analysis are usually overlooked. However, a long-standing body of research (Szabó and Babuška, 2021) has endowed the finite element framework with tools that allow the preservation of topological structures, mimetic discretizations (Castillo and Miranda, 2013) and exact enforcement of boundary conditions, to name a few. At the intersection of finite element exterior calculus (FEEC) (Arnold, 2018; Trask et al., 2022; Actor et al., 2024) and operator learning, conditional Whitney forms (CWFs) (Kinch et al., 2025) have recently emerged as a framework that enables the construction of learnable reduced mixed finite element spaces. Most importantly, due to the mimetic nature of the underlying discretization and the mixed-space approach, CWFs can ensure the preservation of data-driven conservation laws.

Motivated by the sound theoretical foundation and the non-invasive implementation of CWFs, we explore their ability to turn standard operator learning architectures into structure-preserving reduced-order models and, consequently, be employed in challenging learning setups. As preliminary experiments indicated, the equality-constrained optimization problem, as posed in the original formulation (Kinch et al., 2025) and applied in subsequent experiments, leads to trivial forms of conservation, ignoring the actual physics. We provide an intuitive, yet rigorous, interpretation of this phenomenon and highlight the necessity of additive regularization to mitigate it. Equipped with the above insights, we reformulate the optimization problem to enable actual-physics recovery and validate our approach in a suite of four problems of increasing difficulty, characterized by a conservation law. Our main contributions can be summarized as follows.

- **Theoretical Insights.** We provide insights into the simplicity of the conservation constraint in the existing formulation of conditional Whitney forms (CWFs) and demonstrate the necessity of added regularization to reveal their true capacity to achieve ground-truth physics recovery.

- **Physics-aware reformulation.** We reformulate the existing learning problem by adding a data-driven physics recovery component and employ the conditional Whitney forms framework to transform a standard operator-learning architecture into a reduced-order model that precisely preserves conservation laws and enforces boundary conditions.

- **Comprehensive evaluation.** We test the validity of our approach on four problems governed by conservation equations and of increasing difficulty. Empirical evidence shows that our method enables physics recovery while exactly preserving conservation laws and only slightly affecting the expressivity of the original architecture.

## 2 BACKGROUND AND RELATED WORK

**Operator Learning.** Neural operators (Kovachki et al., 2023) have emerged as a powerful alternative to traditional PDE solvers. Early research focused on the generalization of deep learning architectures such as feed-forward and convolution networks in the infinite-dimensional setting and the construction of appropriate bases to express PDE solutions, such as Fourier Neural Operator (Li et al., 2020) and DeepONet (Lu et al., 2021). In addition to attempts to build on these foundational ideas and develop more efficient models (Bonev et al., 2023; Zhu et al., 2023; Kopaničáková and Karniadakis, 2025), motivated by advancements in NLP and Computer Vision, Transformer-based approaches (Kissas et al., 2022; Li et al., 2022; Hao et al., 2023) came to merit attention. Specifically, Wang et al. (2024) provided a unified perspective between operator learning and conditioned neural fields (Xie et al., 2022). The adoption of self-attention mechanisms as the main building block in operator designs and subsequent advances enabled the development of the first scientific foundation models (Hao et al., 2024; Herde et al., 2024; Bodnar et al., 2025). However, physics is still only partially included as a mere inductive bias in such learning pipelines (Li et al., 2024; Zhang et al., 2025), improving but not providing guarantees on the physical realizability of the proposed solutions in most cases.

**Structure-Preserving Scientific Learning.** Recently, a plethora of works have employed components of finite element analysis to enhance the structure-preserving properties of machine learning pipelines. Ouyang et al. (2025) utilized the output of pre-trained neural operators as elements in traditional FEM pipelines. However, due to the presence of learnable elements, the quadrature rule inserts variational crime in the computation of stiffness matrices and prevents conservation. Bouziani and Boullé (2024) proposed a FEM-inspired architecture, in which the preservation of structure is

limited to boundary conditions and a topological inductive bias in the architecture. Furthermore, Bouziani et al. (2024) and (Farsi et al., 2025) presented frameworks that allow the integration of FEM (Ham et al., 2023) and deep learning(Bradbury et al., 2018; Paszke et al., 2019) python libraries in an end-to-end differentiable way, enabling the solution of inverse problems with learnable components and the usage of variational residuals as regularization terms among other utilities. Finally, Sunil and Sills (2024) and Rezaei et al. (2024) proposed alternative PINN (Raissi et al., 2019) formulations, where the training loss and design choices are inspired by finite element analysis. Although these works provide useful insights and interesting directions, to the best of our knowledge, CWFs are the only framework that allows the transformation of deep learning architectures into structure-preserving reduced-order models with only modest computational overhead and minimal architectural intervention.

## 3 LEARNING PROBLEM REFORMULATION

### 3.1 PRELIMINARIES

Inspired by Whitney forms (Lohi and Kettunen, 2021) in FEEC, Actor et al. (2024) showed that any partition of unity ($\{\psi_i(\cdot)\}$ s.t. linearly independent, $\psi_i(\cdot) \geq 0$ and $\sum_i \psi_i(\cdot) = 1$) can be used as the basis for the construction of a family of finite element spaces that constitute a mimetic discretization (Castillo and Miranda, 2013) of the de Rham complex, essentially providing exact discrete analogues of the standard differential operators (e.g. gradient, divergence) and a generalized Stokes theorem.

In the 2D setting, using the finite element spaces:

$$\mathbb{W}^0(\Omega) = \text{span}\{\psi_i^0(\cdot)\}, \quad \mathbb{W}^1(\Omega) = \text{span}\{\psi_{ij}^1(\cdot) = \psi_j^0 \nabla \psi_i^0 - \psi_i^0 \nabla \psi_j^0\},$$

to solve a PDE of the following conservative form:

$$\nabla \cdot (\nabla u + N[u; \phi]) = s, \qquad u = 0 \text{ on } \partial\Omega,$$

where $N(\cdot, \phi)$ denotes a generic flux operator with parametrization $\phi$, mixed Galerkin form seeks $(u, f) \in \mathbb{W}^0(\Omega) \times \mathbb{W}^1(\Omega)$ such that for all $(q, v) \in \mathbb{W}^0(\Omega) \times \mathbb{W}^1(\Omega)$:

$$-(f, \nabla q)_\Omega = (s, q)_\Omega, \qquad (f, v)_\Omega = (\nabla u, v)_\Omega + (N[u; \phi], v)_\Omega.$$

Theorem 3.4 from Kinch et al. (2025) yields an equivalent representation of the above formulation:

$$\delta_0^T M_1 \hat{f} = M_0 \hat{s}, \quad M_1 \hat{f} - M_1 \delta_0 \hat{u} - M_1 \hat{N}[\hat{u}; \phi] = 0, \tag{1}$$

where $\hat{(\cdot)}$ denotes a discretized version of functions or operators, $M_0$, $M_1$ denote the mass matrices associated with $\mathbb{W}^0(\Omega)$, $\mathbb{W}^1(\Omega)$ and $\delta_0$ denotes a generalized incidence matrix between $\mathbb{W}^0(\Omega)$ and $\mathbb{W}^1(\Omega)$ s.t. $(\delta_0)_{ij,i} = -1$, $(\delta_0)_{ij,j} = 1$ and $\delta_0 = 0$ otherwise.

Using the closeness of partitions of unity under convex combinations, the degrees of freedom of a fixed partition of unity (fine-grained partition) can be mapped to the degrees of a reduced one (coarse-grained partition) through a simple multiplication with a learnable column stochastic matrix:

$$W(z; \theta) = \begin{bmatrix} W^{\text{int}}(z; \theta) & 0 \\ 0 & W^{\text{bnd}} \end{bmatrix}. \tag{2}$$

The matrix $W(z; \theta)$ can be inferred from $z$ by any standard regression model with parameters $\theta$ and, essentially, maps boundary to boundary and interior to interior degrees of freedom. As proposed in (Kinch et al., 2025), taken together, (1) and (2) give birth to the following PDE-constrained learning problem in the reduced space, where operator $N$ is replaced by a neural network $\mathcal{NN}$:

$$\min_{\hat{u}_i, \theta, \phi} \quad \sum_i \| \sum_j \hat{u}_{i,j} \cdot \psi_j^0(x; z_i, \theta) - u_{i_{\text{target}}}(x)\|_\Omega^2 \qquad i \in \{1, ....N_{\text{samples}}\}, \quad x \in \Omega,$$

$$\text{subject to} \quad \delta_0^\top M_1(z_i; \theta)(\delta_0 \hat{u}_i + \mathcal{NN}[\hat{u}_i; z_i, \phi]) - M_0(z_i; \theta)\hat{s}(z_i) = 0. \tag{3}$$

For more details on these derivations, we refer the reader to Section 2 of Kinch et al. (2025), Section 3 of Actor et al. (2024) and Appendix A.

### 3.2 TRIVIAL CONSERVATION

As posed in (3), the learning problem is related to the inverse problem of discovering a conservation equation applied to a reduced space of learnable finite elements. However, conservation is achieved in a trivial way, which, essentially, means that the constraint can be omitted, as it does not restrict the search space of candidate solutions $u_i(x)$. Thus, the learning problem can be solved as an unconstrained regression problem in $\mathbb{W}^0(\Omega)$ and the conservation constraint can be satisfied with a standard post-processing step in $\mathbb{W}^1(\Omega)$, as it only really imposes a source-flux outflow balance. Formally, the redundancy of the PDE constraint is summarized in the following proposition.

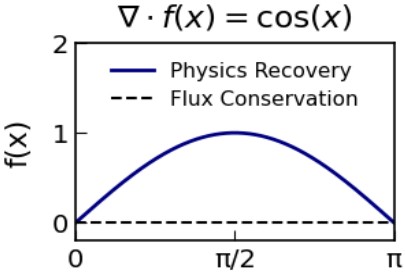

Figure 1: Physics recovery vs. structure preservation.

**Proposition 1.** *Let $u(x) := \sum_i c_i \lambda_i(x)$ s.t. $0 \leq u(x) \leq 1$ and $u(x) = 0$ on $\partial\Omega$, where $\{\lambda_i(x)\}_{i=1}^N$ denotes a fine-grained partition of unity. When function $f(\cdot)$ is not restricted to any specific structure and for any source $s(x)$, there is a trivial reduced partition $\{\psi_j(x)\}$, in which $u(x)$ can be exactly described and satisfies a conservation equation of the form:*

$$\delta_0^\top M_1 \hat{f} = M_0 \hat{s}, \qquad M_1 \hat{f} = M_1 f(\hat{u}), \tag{4}$$

*for an infinite number of vectors $\hat{f}$.*

**Proof:** We define $u(x) := \sum_i c_i \lambda_i(x)$, where $\{\lambda_i(x)\}_{i=1}^N$ denotes a fine-grained partition. We can construct a coarse-grained partition $\{\psi_j(x)\}$ as follows.

$$\psi_0(x) := \sum_i c_i^{\text{int}} \lambda_i^{\text{int}}(x), \quad \psi_1(x) := \sum_i (1 - c_i^{\text{int}}) \lambda_i^{\text{int}}(x), \quad \psi_2(x) := \sum_i \lambda_i^{\text{bnd}}(x).$$

The subsequent transformation matrix $W$ of (2) is :

$$W = \begin{bmatrix} \begin{bmatrix} \mathbf{c}_{int}^T \\ (\mathbf{1} - \mathbf{c}_{int})^T \end{bmatrix} & 0 \\ 0 & \mathbf{1}^T \end{bmatrix},$$

we trivially derive $u(x) = 1 \cdot \psi_0(x) + 0 \cdot \psi_1(x) + 0 \cdot \psi_2(x)$ and denote $\hat{u} := [1, 0, 0]^T$. Equipped with the constructed partition of unity, we can express a conservation equation as in (4). Since everything is known and $f(\cdot)$ is not limited to any specific form, we only have to solve the first system in (4). As we have used $\psi_2(x)$ only to apply the homogeneous boundary conditions, the system admits a reduced form as $(\delta_0^\top)_{1:2,:} M_1 \hat{f} = M_0 \hat{s}$, in which $M_0 \hat{s}$ denotes the projection of the source only on the interior nodes. Since $\delta_0$ is an incidence matrix, $(\delta_0^\top)_{1:2,:}$ takes the following form:

$$(\delta_0^\top)_{1:2,:} = \begin{bmatrix} -1 & -1 & 0 \\ 1 & 0 & -1 \end{bmatrix},$$

which is a full-row rank matrix and $M_1$ is a symmetric positive-definite matrix as the mass matrix of linearly independent finite elements; thus, the system is underdetermined, admitting an infinite number of solutions and concluding the proof.

**Remark 1.** *We can drop the assumptions pertaining to $u(x)$ in Proposition 1 by slight modifications to the proof. We can generalize for any $u$ s.t. $u_{min} \leq u(x) \leq u_{max}$ by setting $\hat{u} := [u_1, u_2]^T$, where $u_1 \geq u_{max}$ and $u_2 \leq u_{min}$ and rescaling $\mathbf{c}_{int}$. Furthermore, we can drop the homogeneous boundary conditions with a decomposition of the fine-grained boundary degrees of freedom, identical to the decomposition used for the interior ones.*

One could assume that increasing the cardinality of the learnable partition of unity would suffice to endow the method with implicit constraints regarding flux reconstruction, similarly to how a finer discretization improves the stability and approximation properties of a standard numerical method. However, we can generalize Proposition 1 for any cardinality of the coarse-grained partition of unity with standard arguments from linear algebra.

**Proposition 2.** *Let $u(x) := \sum_i c_i \psi_i(x)$ s.t. $u(x) = 0$ on $\partial\Omega$, where $\{\psi_i(x)\}_{i=1}^M$ denotes a coarse-grained partition of unity. When function $f(\cdot)$ is not restricted to any specific structure and for any source $s(x)$ with projection $M_0 \hat{s}$ on $\{\psi_i(x)\}_{i=1}^M$, the conservation constraint:*

$$\delta_0^\top M_1 \hat{f} = M_0 \hat{s} \tag{5}$$

*is satisfied for an infinite number of vectors $\hat{f}$ and $null(\delta_0^T M_1) = \binom{M-1}{2}$.*

**Proof:** The full proof is provided in Appendix B. Briefly, we prove the proposition by analyzing the rank of $\delta_0$.

### 3.3 FLUX REGULARIZATION

Adopting a mixed-space approach to express a data-driven generalized conservation law, CWFs are endowed with tools to treat boundary conditions, source-flux outflow balance and other structure-preserving properties in a precise way. The block-diagonal structure of (2) plays a critical role in the preservation of the structure. It allows for the performance of the lift, which is necessary to impose (in-)homogeneous Dirichlet boundary conditions, and for the quantification of the flux outflow, which is necessary for the source-flux outflow balance claims.

However, the findings of Section 3.2 indicate that the lack of any structure in the reconstructed flux $\hat{f}$ leads to the triviality of (3). In practice, this translates into trivial reduced partitions and flux hallucinations, contradicting the initial selection of a mixed-space approach to describe the physics while preserving a conservation law. Furthermore, experimental evidence shows that, in settings where a machine learning model is used for the inference of $W(z; \theta)$, the predictive performance of CWFs is almost identical to that of the original model, hinting at the findings of Section 3.2.

As we emphasize the importance of flux regularization, we propose the reformulation of (3), adding a flux reconstruction term as follows.

$$\min_{\hat{u}_i, \theta, \phi} \quad \sum_i \| \sum_j \hat{u}_{i,j} \psi_j^0(x; z_i, \theta) - u_{i_{\text{target}}}(x) \|_\Omega^2 +$$

$$+ \lambda \sum_i \| \sum_{jk} \hat{f}_{i,jk} \psi_{jk}^1(x; z_i, \theta) - f_{i_{\text{target}}}(x) \|^2, \tag{6}$$

$$\text{subject to} \quad \delta_0^\top M_1(z_i; \theta) \hat{f}_i - M_0(z_i; \theta) \hat{s}(z_i) = 0,$$

$$M_1(z_i; \theta)(\hat{f}_i - \delta_0 \hat{u}_i - \mathcal{NN}(\hat{u}_i; \phi, z_i)) = 0.$$

We adopt the above data-driven approach of flux reconstruction, as learning optimal geometries for the representation of the governing physics is one of the main desiderata of CWFs. To be more precise, the above form of $\hat{f}_i$ contains both a structural assumption and a learnable component, as $\delta_0 \hat{u}_i$ exactly represents the gradient term of flux (diffusion); see Actor et al. (2024), and $\mathcal{NN}(\hat{u}_i; \phi, z_i)$ denotes a learnable nonlinear flux term (convection).

## 4 EXPERIMENTS

We probe the validity of the theoretical insights in Section 3.2 and the effectiveness of the proposed reformulation of Section 3.3 in a suite of four advection-diffusion systems. In addition, we conduct an ablation on how the flux penalty $\lambda$ affects reconstruction for a Poisson problem in which discretization choices have altered the conservative nature of the flux. In the remainder of this section, we will refer to the optimization approach of Section 3.3 as the regularized method, while we will use the term unregularized method for the original formulation.

**Model Setup.** We use a CViT (Wang et al., 2024) model for the inference of learnable Whitney forms. CViT combines a vision transformer encoder, a grid-based coordinate embedding and a query-wise cross-attention mechanism with state-of-the-art results in a suite of temporal nonlinear PDEs. We add a cross-attention module with learnable input queries to extract a low-dimensional conditioning variable for the $\mathcal{NN}$ in (6). In problems with global parameters (Experiments 4.3 and 4.4) of the underpinning PDE, we use these parameters as an extra conditioning token in the self-attention layers of the encoder. Regarding CWFs, we select a $Q1$ basis on a $128 \times 128$ quadrilateral mesh as the fine-grained partition of unity and select 8 learnable basis functions for the coarse-grained partition. Furthermore, we replace the attention-based flux model of Kinch et al. (2025) with a small MLP network, since we observed that larger models made training unstable and significantly slower, when the flux reconstruction term was added.

**Experimental Setup.** Advection-diffusion equations are used to describe a plethora of phenomena encountered in fields such as environmental science, oceanography and thermal sciences. In their

stationary version, they are described by the following conservation equation.

$$\nabla \cdot (v(x)\nabla u(x) + a(x)u(x)) = s(x), \tag{7}$$

where $u(x)$ denotes the distribution of a quantity, $v(x)$ denotes the diffusivity tensor (or viscosity), $a(x)$ denotes the velocity vector and $s(x)$ denotes the source. For all subsequent experiments, we assume canonical domains $\Omega := [0,1]^n$. We create learning problems of increasing difficulty by a range of variations in the parameters of (7), which we present in the following sections.

## 4.1 1D ADVECTION DIFFUSION

We generate the data according to the following PDE:

$$\frac{d^2u}{dx^2}(x) - \frac{1}{\epsilon}\frac{du}{dx}(x) = 0,$$

where $u(0) = 1$ and $u(1) = 0$ and $\epsilon$ denotes the inverse of the Péclet number. We create different instantiations of the PDE by sampling $\epsilon$ as $\sim \mathcal{N}(0.5, 0.2^2)$ clipped in $[0.1, 0.9]$. We use 90 samples to train and 10 samples to evaluate. We convert the Péclet number into a token $z$ with a simple MLP, which we use as input to the decoding part of a CViT to infer the coarse-grained partition of unity.

We use this pedagogical example to demonstrate that CWFs without regularization do not recover the actual flux, even in toy problems, although they achieve exact conservation. The results are summarized in Table 1. As expected, the distribution reconstruction deteriorates with the addition of the flux reconstruction term, since a single partition of unity should accommodate the reconstruction for both distribution $u$ and flux $f$ via the Whitney 0- and 1-forms construction. In Figure 2, we see that the unregularized method resorts to trivial representations of the learnable part of the flux, adjusting it near boundaries to achieve the source-outflow balance encoded in the equality constraint.

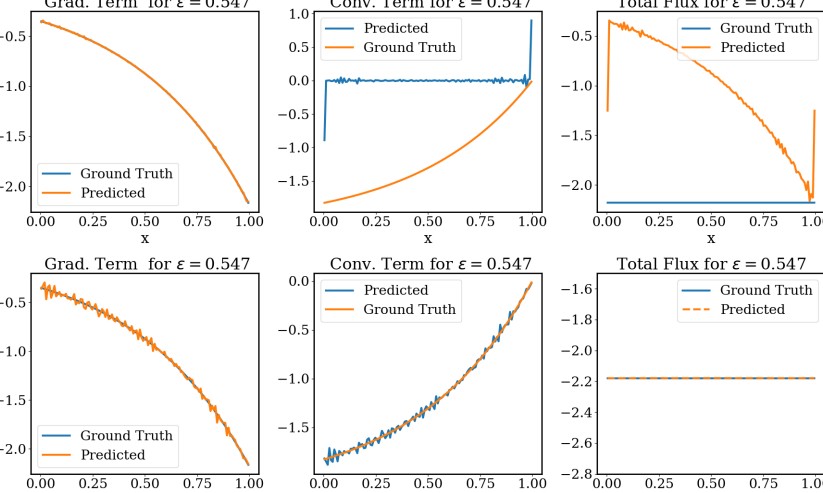

Figure 2: *1D Advection Diffusion.* Representative flux reconstruction for the unregularized (top) and the regularized method (bottom). In the unregularized case, steep adjustments balance out flux inflow-outflow to match the absence of a source term.

## 4.2 2D INHOMOGENEOUS ADVECTION DIFFUSION

Adopting the idea of the Darcy flow dataset from (Takamoto et al., 2022), we create different realizations of (7) by dividing $\Omega$ into 2 subdomains $\Omega_1$ and $\Omega_2$, as shown in Figure 3, and applying different velocities within these domains. The velocity field $a(x, y)$ is defined as follows.

$$a(x, y) = \begin{cases} -5 \cdot \mathbf{1}^T & \text{if } (x,y) \in \Omega_1 \\ -0.1 \cdot \mathbf{1}^T & \text{if } (x,y) \in \Omega_2 \end{cases}.$$

Furthermore, we set $s(x, y) = 20 \sin(\pi x) \cos(\pi y)$ and $v = 1$ for the entire dataset. We divide a dataset of $10,000$ samples into $9,500$ training and $500$ testing samples.

We summarize the results in Table 1. Surprisingly, we observe that the regularized method does not affect the distribution reconstruction, while it drastically improves the flux reconstruction. This may indicate that distribution reconstruction is limited by learning properties (dataset/model size, etc.) and not by an inherent inability of the learned Whitney forms to reconstruct both distribution $u$ and flux $f$ as previously. Once again, we see that the unregularized method learns uniform fluxes, achieving flux balance through adjustments near the boundaries (see Figure 4).

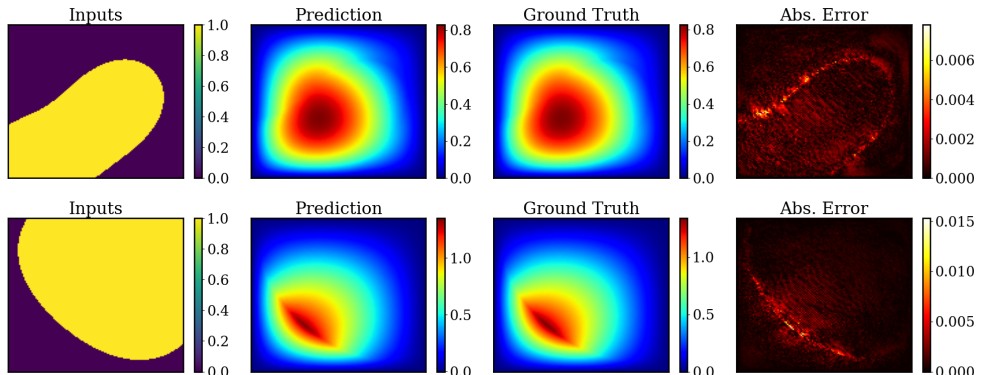

Figure 3: *2D Inhomogeneous Advection Diffusion.* Representative domains, predictions and point-wise errors for the regularized method.

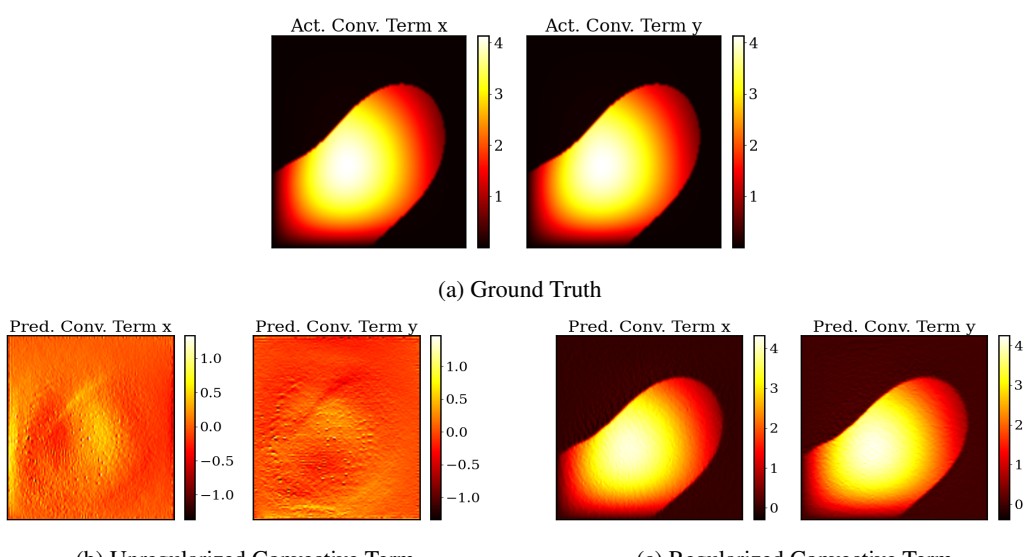

(a) Ground Truth

(b) Unregularized Convective Term

(c) Regularized Convective Term

Figure 4: *2D Inhomogeneous Advection Diffusion.* Representative flux reconstruction for both methods. Similarly to the *1D Advection Diffusion*, we observe that the regularized convective term is coarser than the ground truth, although it captures the general discontinuous form. This is reasonable as we minimize the reconstruction error of the total flux. However, we pick to showcase the convective term, as this is the learnable part of the flux and pictures a stark contrast to the unregularized method. Added visualization pertaining to flux reconstruction is presented in Appendix D.

### 4.3   2D ADVECTION DIFFUSION

In this experiment, we create different realizations of (7) by sampling random velocity vectors with a Euclidean norm in the range of $[0, 20]$ and source terms of variable sparsity as in Subramanian

et al. (2023) (see Figure 5) and fix viscosity $v = 1$ for the entire dataset. We divide the dataset into a training set of $32,768$ samples and a test set of $4,096$ samples. Once again, the regularized method achieves flux reconstruction, while only slightly harming distribution reconstruction (see Table 1).

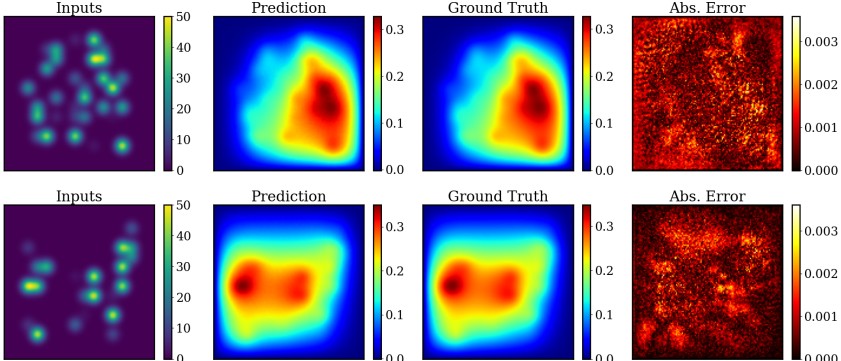

Figure 5: *2D Advection Diffusion.* Predictions and point-wise errors for the regularized method.

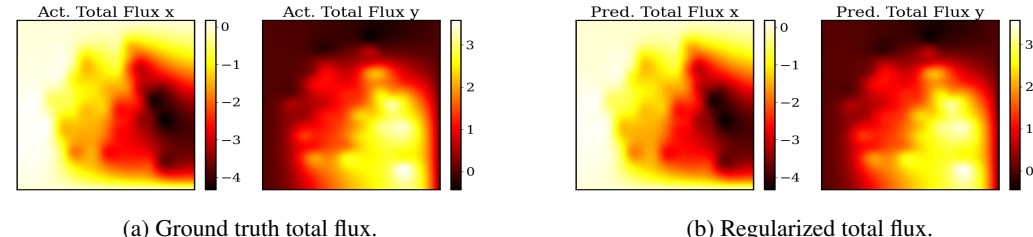

(a) Ground truth total flux.         (b) Regularized total flux.

Figure 6: *2D Advection Diffusion.* Reconstructed and ground truth total flux.

Table 1: MSE for distribution and flux reconstruction in Experiments 4.1-4.3 with the unregularized (first row) and the regularized method with $\lambda = 0.01$ (second row).

| Method | 1D Adv.-Dif. | | 2D Adv.-Dif. Inhom. | | 2D Adv.-Dif. | |
|---|---|---|---|---|---|---|
| | Dist. Error | Flux Error | Dist. Error | Flux Error | Dist. Error | Flux Error |
| Unreg. | $8.2 \cdot 10^{-10}$ | $2.4 \cdot 10^{0}$ | $1.5 \cdot 10^{-6}$ | $3.0 \cdot 10^{0}$ | $4.5 \cdot 10^{-7}$ | $7.1 \cdot 10^{0}$ |
| Reg. | $7.6 \cdot 10^{-8}$ | $3.6 \cdot 10^{-7}$ | $1.5 \cdot 10^{-6}$ | $1.6 \cdot 10^{-3}$ | $1.2 \cdot 10^{-6}$ | $5.8 \cdot 10^{-4}$ |

### 4.4 2D Anisotropic Poisson Equation

So far, we have been working with standard isotropic diffusion terms. However, since Whitney forms allow for the exact expression of the diffusion term, we can vary the diffusion tensor without sacrificing the capacity of Whitney forms to exactly express the diffusion term and the subsequent mass matrices. Specifically, we set the diffusivity tensor $v := R^T D R$, where $R = rot(\theta)$ denotes a rotation matrix that controls the direction of diffusion with $\theta \sim \mathcal{U}(0, 2\pi)$ and $D = diag(1, e)$ denotes a diagonal matrix that controls the degree of diffusion anisotropy with $e \sim \mathcal{U}(1, 5)$. Only a slight modification should be made to the stiffness matrix of the diffusion term in

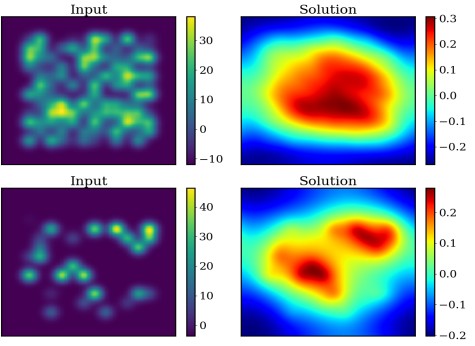

Figure 7: *Anisotropic Poisson Equation.*

(1). To further reinforce the variability of the solutions, we sample the source term as in Experiment 4.3 and apply periodic boundary conditions. Representative samples are presented in Figure 7.

We divide the dataset into a training set of $32,768$ samples and a test set of $4,096$ samples. For further evaluation of the reconstruction losses achieved by different flux penalties $\lambda$, we mention that the mean absolute value of $u$ equals $0.235$ and the mean absolute value of flux $f$ equals $0.865$ as a reference. We sum up the results of our experiments in Table 2. Interestingly enough, setting $\lambda = 0.001$, we achieve

Table 2: Ablation on variable flux penalty $\lambda$.

| Method | Dist. Loss | Flux Loss |
|---|---|---|
| Unregularized | $8.2 \cdot 10^{-7}$ | $3.1 \cdot 10^{-1}$ |
| Flux Pen. 0.1 | $1.2 \cdot 10^{-5}$ | $4.2 \cdot 10^{-4}$ |
| Flux Pen. 0.01 | $1.8 \cdot 10^{-5}$ | $2.2 \cdot 10^{-3}$ |
| Flux Pen. 0.001 | $1.8 \cdot 10^{-6}$ | $5.3 \cdot 10^{-4}$ |
| Flux Pen. 0.0001 | $1.5 \cdot 10^{-6}$ | $9.1 \cdot 10^{-3}$ |

better flux reconstruction than larger penalties, demonstrating the importance of distribution reconstruction in flux reconstruction of diffusion-dominated problems.

## 5 DISCUSSION

**Summary.** This work sheds light on the recently proposed framework of conditional Whitney forms (Kinch et al., 2025). Building upon foundational concepts of finite element exterior calculus, CWFs introduce a learning paradigm that operates in mixed spaces of finite elements. Thus, it is equipped with a strong theoretical machinery to embed structure-preserving properties, such as conservation laws and boundary conditions, into learnable reduced-order models.

However, the solution of the learning problem, as originally proposed, leads to trivial representations of both geometry and physics. We essentially show that, when no specific structure of the physics is assumed, the constrained-optimization problem is equivalent to an unconstrained regression task, followed by a standard post-processing step. We also stress the necessity of physics regularization and propose a data-driven approach to recover the ground-truth physics. Finally, we test the validity of our theoretical insights and evaluate the performance of CWFs in the reformulated learning problem in a range of four advection-diffusion systems of increasing difficulty. The experimental results support our theoretical claims, while CWFs exhibit notable adaptability to the new challenges presented.

**Further discussion.** As mixed FEM-ML approaches have received increasing attention (Rezaei et al., 2024; Bouziani and Boullé, 2024; Ouyang et al., 2025; Farsi et al., 2025), we acknowledge conditional Whitney forms as a very promising direction for the design of reliable machine learning models with guaranteed physical realizability. Yet, we reckon that they have only been employed in learning problems, where the structure-preserving properties can be achieved predominantly due to the selection of a mixed-space approach and do not mingle with the task of actual-physics recovery, as a mixed-space approach would suggest. Since we are the first to test the performance of CWFs in such tasks, we hope that our findings will motivate further work on this framework. Future work could provide better insights into both experimental and theoretical aspects of CWFs, such as performance in highly nonlinear settings, the trade-off between physics recovery and stability, and efficient implementations.

Physics recovery, particularly, poses significant new challenges for CWFs. First, a partition of unity that accommodates both distribution and flux reconstruction has to be learned for each sample. However, as problems leave the domain of diffusion-dominated systems, this task may present difficulties generally pertaining to the FEEC treatment of nonlinear systems or the reduced-order approach. In addition, it is not clear how an accurate recovery of highly nonlinear physics affects the well-posedness of the equality constraint; the core piece of structure preservation. We present the example of a not so well-conditioned equality constraint and discuss it further in Appendix C. Finally, the addition of a flux-reconstruction term to the loss function may significantly affect specific design and optimization choices. For example, it favors the selection of simple models to represent the learnable flux term for both stability and computational efficiency reasons. Thus, new experimental evidence is needed on the optimal implementation of CWFs.

ETHICS STATEMENT

This work proposes physics-informed learning techniques that can accelerate the modeling of complex physical systems and enable more efficient and reliable simulation tools. Our methods rely solely on synthetic or domain-agnostic data, and we commit to transparency by releasing code and detailed methodological descriptions to support reproducibility. Although we do not anticipate specific negative impacts from this work, we recognize that, as with any powerful predictive tool, there is potential for misuse; we therefore encourage the research community to carefully consider ethical implications and potential dual-use scenarios when applying these technologies in sensitive domains.

REPRODUCIBILITY STATEMENT

A detailed description of the data generation process and the experimental setup that was followed is provided in Section 4. Additional information pertaining to method implementation, model architecture and data setup can be found in Appendix A. The code used to carry out the experiments is provided as a supplementary material for this submission.

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

# A    Implementation Details

**Whitney Forms.**    A significant benefit of conditional Whitney forms is the non-invasive computation of the mass matrices $M_0^c$, $M_1^c$ of the coarse-grained partition of unity from the mass matrices $M_0^f$, $M_1^f$ of the fine-grained partition of unity. We recall that the mass matrices are generally computed as follows:

$$(M_0)_{i,j} = (\psi_i^0, \psi_j^0)_\Omega,$$
$$(M_1)_{ij,kl} = (\psi_{ij}^1, \psi_{kl}^1)_\Omega,$$

where $(\cdot, \cdot)_\Omega$ denotes the $L^2$-inner product of the functions $f : \Omega \to \mathbb{R}$ and the functions $g : \Omega \to \mathbb{R}^2$ correspondingly. The mass matrices $M_0^c$, $M_1^f$ can be computed as follows:

$$M_0^c = W M_0^f W^T,$$
$$M_1^c = (W \otimes W) M_1^f (W \otimes W)^T.$$

Since the fine-grained partition consists of $Q_1$ elements on a quadrilateral mesh, we compute the mass matrices through a sparse, diagonalized implementation of the assembly algorithm proposed in Appendix A.3 of Actor et al. (2024). The interested reader can find more about this implementation in the code uploaded.

**Equality-Constrained Optimization.**    We adopt the optimization scheme proposed in Section 3 of Kinch et al. (2025) to ensure that the equality constraint is met throughout the training enforcing the preservation of structure, strictly and independently of the predictive performance of the model. In addition, we use the mean squared error of the distribution reconstruction on the nodes of the rectangular mesh and the mean squared error of the flux reconstruction on the midpoint of the rectangular cells to train and evaluate our model. We replace the Shampoo optimizer (Gupta et al., 2018) with a typical AdamW one (which seems to be working fine) for computational efficiency reasons, as the model size, the data dimensionality and the flux reconstruction setup increase the computational load of CWFs in our setup. As the flux reconstruction term integrates the governing physics into the learning problem, the optimality of the employed optimization schemes and their correlation to the corresponding categories of physics needs to be investigated. More discussion on this topic is provided in Appendix C.

**CViT.**    Our main interventions in the CViT architecture are described in the model setup section of the main paper. We use similar hyperparameters for all our 2D experiments. We train the models for 250,000 steps with batch size 16 and optimize with AdamW using an exponential learning rate scheduler. More details on the exact hyperparameters used can be found in the configuration files provided for each experiment in the code uploaded.

**Data Setup.**    We provide an analytic description of the data generation and training/validation setup in the main body of the paper. The 1D problem admits an analytical solution, while we use scikit-fem (Gustafsson and McBain, 2020) to generate solutions for the 2D experiments. For the problem with periodic boundary conditions, we use the code provided by $Subramanian\,et\,al.$ (2023) on `https://github.com/ShashankSubramanian/neuraloperators-TL-scaling/tree/main/utils`. Any details regarding the data generation procedure, which were possibly omitted here, can be found in the supplementary code.

**Computational Cost.**    All experiments were performed on two NVIDIA H200 Tensor Core GPUs. Training times for 250,000 steps and a 16 batch size were 20 hours for the 2D problems. Without flux reconstruction, training times were around 9 hours. This is an expected increase in computational times, as flux reconstruction implies the computation of Whitney 1-forms, whose number is quadratic to the size of the partition of unity. When flux reconstruction is of lesser value, the increased computational cost can be mitigated by sparser evaluations of the flux, etc.

# B  PROOF OF PROPOSITION 2

**Proof:**  As a symmetric positive definite matrix, $M_1$ is a bijective linear map and does not alter the rank-nullity properties of $\delta_0$. Therefore, we only have to analyze $\delta_0^T$ or equivalently $\delta_0$, since:

$$rank(\delta_0^T) = rank(\delta_0).$$

We recall that $\delta_0$ denotes a generalized incidence matrix between elements of Whitney-0 forms $\mathbb{W}^0(\Omega)$ and Whitney 1-forms $\mathbb{W}^1(\Omega)$ that mimics the incidence matrix between the nodes and the edges nodes of a complete undirected graph. Briefly, $\delta_0$ takes the following form:

$$(\delta_0)_{ij,k} = \begin{cases} 1, & \text{if } k = j, \\ -1, & \text{if } k = i, \\ 0, & \text{otherwise} \end{cases},$$

for $i < j$. Therefore, $\delta_0 \in \mathbb{R}^{\binom{M}{2} \times M}$, where M denotes the cardinality of the coarse-grained partition of unity. We can show that the null space of $\delta_0$ is spanned by $\mathbf{1}^T$. Let $x \in \mathbb{R}^M$ s.t. $x = [x_1, ..., x_m]^T$. Then

$$\delta_0 x = 0 \implies ||\delta_0 x|| = 0 \implies x^T \delta_0^T \delta_0 x = 0 \implies \sum_{i<j} (x_i - x_j)^2 = 0 \implies x_1 = ... = x_M.$$

(8)

From the rank-nullity theorem, we have the following.

$$rank(\delta_0) + null(\delta_0) = M,$$

which yields $rank(\delta_0) = M - 1$. Now, we need to recall that, since we use the last node to prescribe the homogeneous boundary conditions, we should practically study the rank of $(\delta_0^T)_{1:M-1,:}$ or equivalently, the rank of $(\delta_0)_{:,1:M-1}$, which is denoted as $\delta_c$. The rank-nullity theorem yields

$$rank(\delta_c) = M - 1 - null(\delta_c).$$

Hence, showing that $\mathbb{N}(\delta_c) = \{0\}$, where $\mathbb{N}$ denotes the null space, suffices to conclude that $rank(\delta_c) = M - 1$. Let $y = [y_1, ..., y_{M-1}]^T$ s.t. $\delta_c y = 0$. This implies that $x = [y, 0]^T$ is a solution to $\delta_0 x = 0$, which makes $y = 0$. Hence, we conclude that:

$$\mathbb{N}(\delta_c) = \{0\} \implies rank(\delta_c) = M - 1.$$

Thus, $rank(\delta_c^T) = rank(\delta_c) = M - 1$ and the equation $\delta_c^\top M_1 \hat{f} = M_0 \hat{s}$ admits infinite solutions $\hat{f}$ for any $\hat{s}$. In addition, the rank-nullity theorem yields the following:

$$null(d_c^T M_1) = M(M - 1)/2 - rank(d_c^T M_1) = M(M - 1)/2 - (M - 1) = \binom{M - 1}{2}.$$

## C  Nonlinear Physics

As already addressed in the main body of our work, learning with conditional Whitney forms in the form of Kinch et al. (2025) is identical to solving a typical regression problem in the degrees of freedom of a finite element space. In addition, since no treatment of the governing physics is taken, no characterizations can be given to the performance of CWFs in taxonomies of physics domains, such as diffusion- or advection-dominated, linear or nonlinear, etc. Experimentally, this translates into a predictive performance identical to the predictive performance of the vanilla data-driven model used for the inference of the partitions of unity.

Proposition 1 shows that any tuple of distribution and source $(u(x), s(x))$ can give birth to a trivial conservation equation that admits an infinite number of solutions for the flux coefficients and completely ignores the actual governing physics. Even if one does not adopt the proposed simplified reformulation of Proposition 1 and sticks to the original formulation of the learning problem, extensive experiments show that CWFs can replicate the predictive performance of the vanilla model without difficulty, but also without any generalization benefit as a constrained approach should. As implied by our work, CWFs can successfully perform regression tasks that are not associated with conservative physics or any physics in general.

However, once the recovery of governing physics is incorporated into the learning task, Whitney forms face significant new challenges. First, the approach of Proposition 1 is no longer valid, since the CWFs must learn a partition of unity that simultaneously supports the reconstruction of both distribution $u$ via Whitney 0-forms ($\mathbb{W}^0(\Omega)$) and flux $f$ via Whitney 1-forms ($\mathbb{W}^1(\Omega)$). Moreover, the coefficients of the distribution and the flux in their respective bases, $\mathbb{W}^0(\Omega)$ and $\mathbb{W}^1(\Omega)$, must be consistently linked through a strict, learnable conservation equation posed in a reduced mixed space (also learnable). As systems move beyond the domain of linear diffusion-dominated PDEs, overcoming these difficulties is far from trivial.

In particular, for convective systems, accurate flux reconstruction often drives the learnable conservation equation into ill-posed regimes. In practice, this translates into the conservation equation ceasing to be solvable (or solvable under conditions, such as the solver used), thereby threatening to derail the training process. This behavior is amply demonstrated in the (anisotropic) advection-diffusion setup of Subramanian et al. (2023). Specifically, reflecting the data generation process presented in that work, we generate PDE coefficients with an advection-to-diffusion ratio distributed as $\sim \mathcal{U}(0.2, 1)$, corresponding to Péclet numbers up to 50. We only modify the boundary conditions from periodic to homogeneous. Finally, the generated dataset follows the typical $32,768$-$4,096$ split of Experiments 4.3 and 4.4.

We observe that larger flux reconstruction penalties push flux reconstruction through error thresholds that destabilize the learning process via the possible ill-posedness of the equality constraint (see Figure 8). We contrast these learning curves with the respective ones from Experiment 4.3, in which Péclet numbers take values of up to 20 (see Figure 9). In conclusion, this experiment clearly demonstrates that the performance of CWFs is not affected by the governing physics, when no accurate flux is required. However, as the convective term dominates the system, flux reconstruction infuses significant difficulty into the learning task.

Although there is a correlation between strong convection and instability of the CWFs, it is not clear if it pertains to the general formulation (i.e. as a conservation constrained optimization problem) or specific implementation aspects, as the optimization method employed. An interesting direction of future research could investigate how quasi-second-order optimization algorithms may affect the stability of CWFs. In Kinch et al. (2025), they have already used Shampoo, a quasi-second-order optimization method. However, due to the use of the original formulation, no concrete evidence is provided regarding its superiority over AdamW. Our experiments show that AdamW does not face any stability issue too, when no flux regularization is enforced, and therefore, the use of other optimization schemes does not seem to be a necessity. In the reformulated setting proposed here, a second-order approach may indeed impact the training dynamics. Wang et al. (2025) provide an insightful analysis of how second-order approaches affect the performance of another category of physics-informed models, the PINNs.

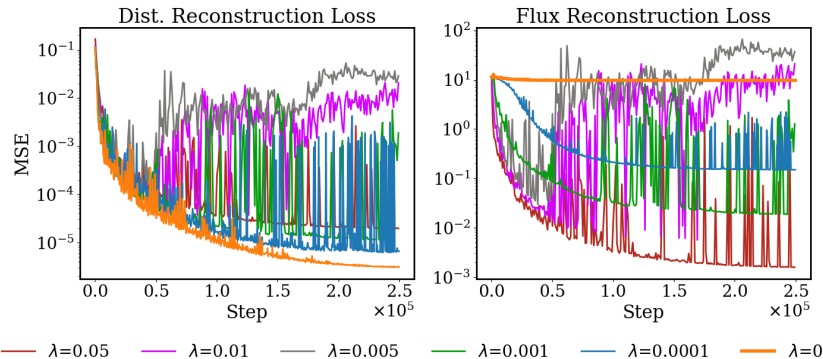

Figure 8: *Highly Convective System.* Distribution and flux reconstruction loss for a range of flux penalties $\lambda$. We observe that for flux penalties $\lambda$ larger than 0.0001 the training becomes unstable. The spikes usually denote an inability to solve the equality constraint (see Figure 10).

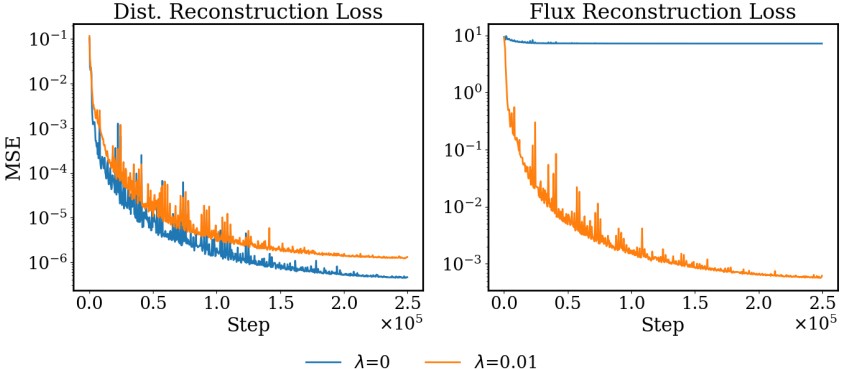

Figure 9: *Moderately Convective System.* Distribution and flux reconstruction validation loss for both methods in Experiment 4.3.

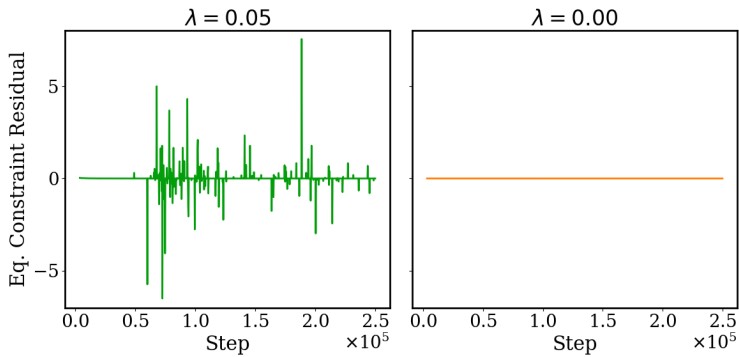

Figure 10: *Highly Convective System.* Equality constraint residual for flux reconstruction penalty $\lambda = 0.05$ and $\lambda = 0.00$ (unregularized). One observes that the equality constraint is satisfied to machine precision throughout the entire training for the unregularized method. On the other side, the model starts struggling with the equality constraint in the range of 50,000-100,000 training steps, when flux regularization is added. This phenomenon may indicate that as the reconstruction and the representation of physics become more accurate, the conservation equation is driven to regimes, where it is not well-conditioned.

## D    SUPPLEMENTARY VISUALIZATIONS

We provide additional visualization associated with the experiments in the main body of the article. We mainly emphasize the reconstruction of the different flux terms.

**1D Advection Diffusion**    For completeness, we present representative field predictions and point-wise errors achieved by the regularized method in Experiment 4.1.

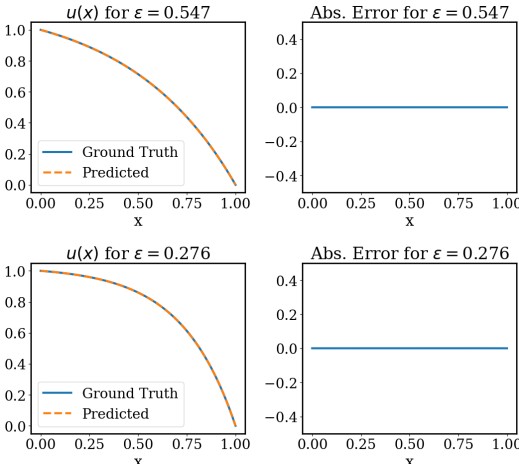

Figure 11: *1D Advection Diffusion.* Representative field predictions and point-wise errors with the regularized method.

**2D Inhomogeneous Advection Diffusion**    We below present gradient and total flux reconstruction for both the unregularized and regularized methods. As the gradient term of the flux is analytic in Whitney forms construction, its reconstruction quality depends only on the reconstruction quality of the distribution $u$. Therefore, we observe that the regularized method gives a slightly coarser representation compared to the unregularized one. However, the total predicted flux is not only able to accurately capture the shapes appearing in the ground truth but also eliminates the coarseness observed in both gradient and convective terms reconstruction.

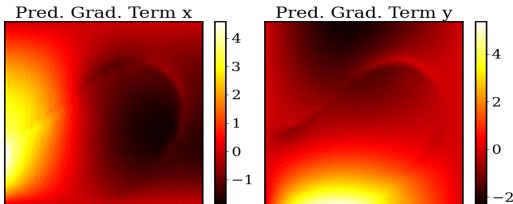

(a) Ground truth gradient term.

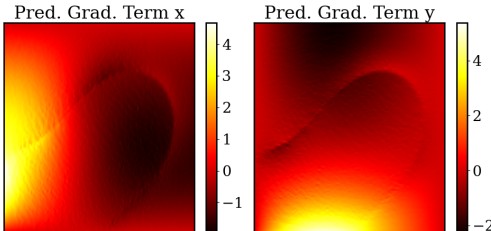

(b) Unregularized gradient term.

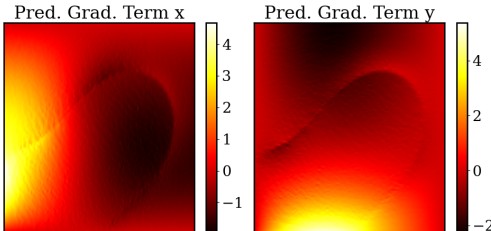

(c) Regularized gradient term.

Figure 12: *2D Inhomogeneous Advection Diffusion.* Representative gradient reconstruction for both methods.

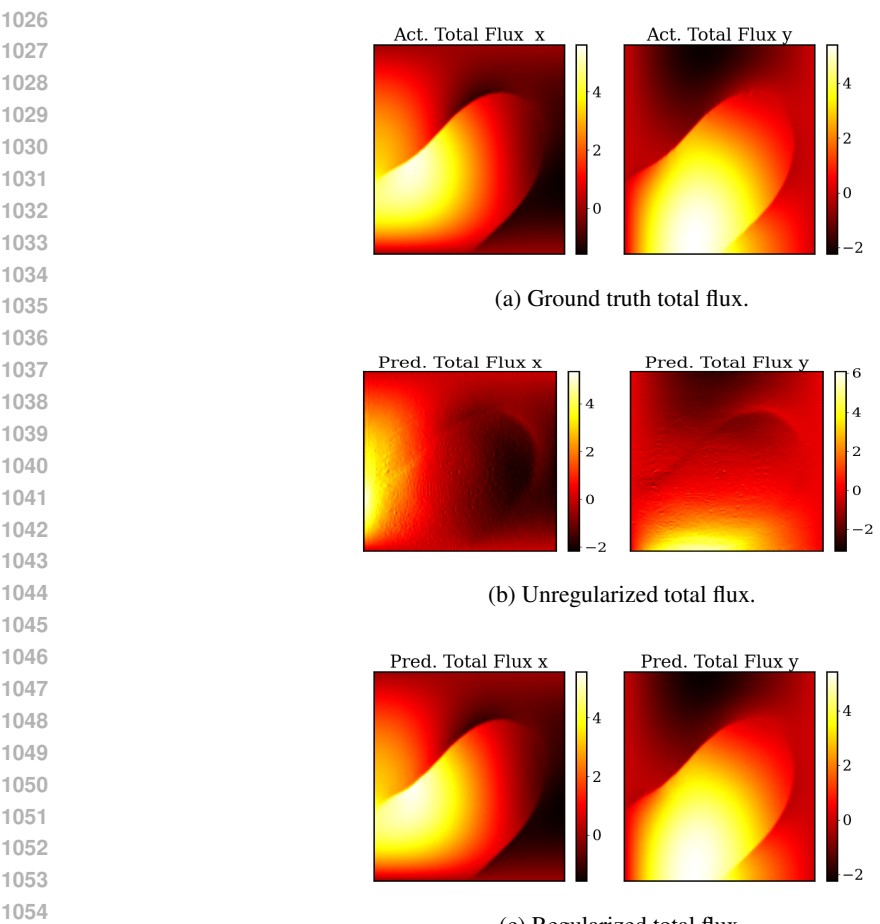

(a) Ground truth total flux.

(b) Unregularized total flux.

(c) Regularized total flux.

Figure 13: *2D Inhomogeneous Advection Diffusion.* Representative reconstruction of the total flux for both methods.

**2D Advection Diffusion** Consistent with the conclusion of Experiment 4.2, the regularized method provides an accurate and smooth representation of the ground-truth flux.

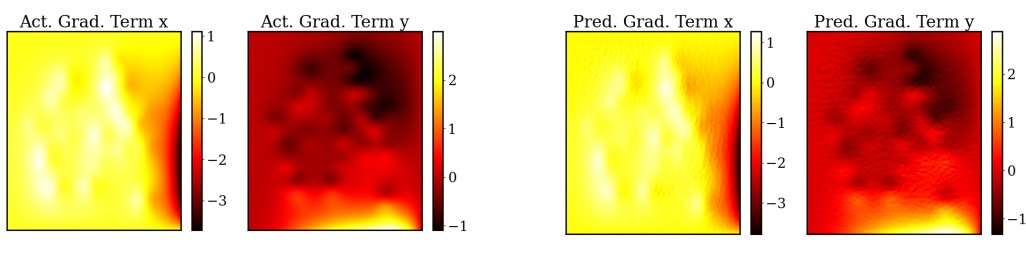

(a) Ground truth gradient term.               (b) Regularized gradient term.

Figure 14: *2D Advection Diffusion.* Reconstructed and ground truth gradient flux terms.

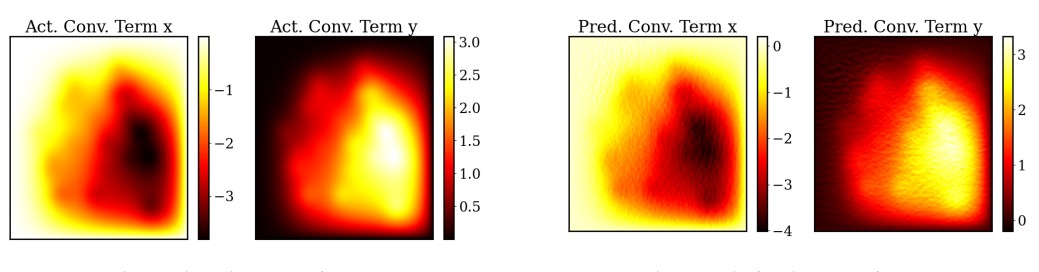

(a) Ground truth convective term.             (b) Regularized convective term.

Figure 15: *2D Advection Diffusion.* Reconstructed and ground truth convective flux term.

