# OpenReview forum: "Revisiting Conditional Whitney Forms: From Structure Preservation to Physics Recovery"
_ICLR.cc/2026/Conference — ICLR 2026 Conference Withdrawn Submission_

### Official Review · Reviewer_bHgX · 2025-10-25

**Soundness:** 3
**Presentation:** 3
**Contribution:** 2
**Rating:** 2
**Confidence:** 2

**Summary:**

This paper revisits Conditional Whitney Forms (CWFs)—a recent framework combining finite-element exterior calculus (FEEC) with neural operator learning—to analyze why prior formulations only trivially satisfy conservation laws without recovering the governing physics. The authors theoretically prove that the original equality-constrained optimization admits infinitely many flux solutions, leading to “formal” but physically meaningless conservation. They then propose a flux-regularized reformulation that introduces a data-driven physics-recovery term and evaluate it on several synthetic advection–diffusion and Poisson systems. Empirically, the regularized model recovers realistic flux fields while preserving structure, confirming the theoretical diagnosis.

**Strengths:**

1. The paper rigorously shows why existing CWFs reduce to unconstrained regression, explaining the phenomenon of trivial conservation through rank-deficiency analysis (Propositions 1–2). This is not really surprising, but it is good for the authors to point out explicitly.

2. Adds an interpretable flux-regularization term that bridges finite-element constraints and learnable operator spaces, linking structure preservation and physics recovery. But this comes with limitations, listed in the "Weaknesses" section.

3. Across 1D and 2D PDEs, the regularized method yields orders-of-magnitude lower flux error while retaining distribution accuracy.

**Weaknesses:**

1. Dependence on ground-truth flux (major limitation!)
The proposed flux-reconstruction loss requires known $f_{true}$. In real scientific-ML settings, fluxes are latent. We typically only observe scalar fields. Thus, the approach is only feasible in synthetic or diagnostic experiments, not in realistic PDE inference or discovery tasks.

2. Limited methodological novelty
The regularization essentially adds a supervised flux-matching term, a straightforward and expected extension of existing physics-informed or operator-learning frameworks. The conceptual leap beyond prior PINN/PINO-style constraints is small.

3. Expected improvement
Since the loss directly penalizes flux error, the performance gain is tautological rather than emergent. The results confirm that additional supervision helps, but do not reveal new modeling behavior.

4. Narrow experimental scope
All tests involve simple 1D–2D advection–diffusion or Poisson problems with synthetic data and fixed meshes. The method’s stability and benefit for nonlinear, convection-dominated, or real-world PDEs remain untested.

5. Limited practical deployability
The computational cost roughly doubles due to flux evaluation, and no clear path is given for extending the approach to unobserved-flux or high-dimensional cases. Without an unsupervised or self-consistent flux-recovery variant, the method’s real impact is theoretical rather than applied.

**Questions:**

1. Can the proposed formulation be adapted to settings where flux is unobserved—e.g., via PDE residuals, energy-based terms, or self-consistent latent-flux estimation?

2. How sensitive is training to the flux penalty and the choice of optimizer (AdamW vs. Shampoo)?

3. Could the flux regularization be interpreted as enforcing an energy-minimization principle or weak form of the PDE?

4. How does the approach scale to 3D problems or complex geometries, where flux computation becomes quadratic in mesh size?

5. Beyond advection–diffusion, does the method extend to nonlinear or coupled PDE systems (e.g., Navier–Stokes)?

---

### Official Review · Reviewer_ergg · 2025-10-30

**Soundness:** 2
**Presentation:** 3
**Contribution:** 2
**Rating:** 4
**Confidence:** 3

**Summary:**

This paper provides a critical analysis and enhancement of the Conditional Whitney Forms (CWFs) framework, which integrates finite element exterior calculus with operator learning. The authors identify a fundamental limitation in the original CWF formulation: the conservation constraint can be satisfied trivially without actually recovering the underlying physics, essentially reducing to an unconstrained regression problem. Through theoretical analysis, they demonstrate that without additional structure, the PDE constraint admits infinite flux solutions. To address this, they propose a reformulated optimization problem with flux regularization that enables true physics recovery while maintaining structure-preserving properties. The method is evaluated on four progressively challenging advection-diffusion problems, showing significant improvements in flux reconstruction with only minor degradation in field prediction accuracy.

**Strengths:**

1.The paper provides deep theoretical insights into why the original CWF formulation fails to recover physics, with formal proofs demonstrating the trivial satisfaction of conservation constraints. This represents a significant contribution to understanding this emerging framework.

2.The proposed flux regularization approach is both elegant and effective, addressing the identified theoretical limitation while maintaining the structure-preserving benefits of CWFs.

**Weaknesses:**

1.The flux reconstruction term significantly increases training time (from ~9 to ~20 hours), which may limit practical applicability, especially since this cost scales quadratically with partition size.

2.The replacement of attention-based flux models with simpler MLPs due to stability issues raises questions about whether the full expressive power of modern architectures can be leveraged.

3.While advection-diffusion problems are well-chosen, evaluation on other PDE families (e.g., wave equations, Stokes flow) would strengthen the claims of general applicability.

**Questions:**

1.You mention replacing the attention-based flux model with an MLP due to training instability. Could this stability issue be addressed through different normalization techniques or optimization strategies, potentially recovering the benefits of more expressive architectures?

2.The quadratic scaling of computational cost with partition size is concerning. Are there approximation techniques or sparse evaluation strategies that could make this approach more practical for large-scale problems?

3.How would your approach handle systems with multiple coupled conservation laws (e.g., thermo-fluid dynamics with energy and mass conservation)? Would the flux regularization need to be extended?

4.The flux penalty parameter λ appears crucial yet problem-dependent. Is there a principled way to set this parameter, or could it be learned adaptively during training?

5.For time-dependent problems, how does the flux regularization affect error accumulation in multi-step rollouts? Does improved instantaneous physics recovery translate to better long-term stability?

6.Your approach relies on a fixed fine-grained mesh for the underlying partition of unity. How sensitive are the results to the choice of this base discretization, and could adaptive mesh refinement be incorporated?

---

### Official Review · Reviewer_W241 · 2025-10-31

**Soundness:** 2
**Presentation:** 2
**Contribution:** 2
**Rating:** 4
**Confidence:** 4

**Summary:**

This paper revisits the framework of Conditional Whitney Forms (CWFs). The authors argue that the original CWF formulation leads to trivial conservation laws and propose a flux-regularized reformulation to enable true physics recovery. They propose a solution by reformulating the learning problem to include a flux reconstruction term as a regularizer. The theoretical claims are supported by two propositions and experimental validation on a series of advection-diffusion problems.

**Strengths:**

- Timely Topic: The work addresses a relevant and emerging topic at the intersection of Finite Element Exterior Calculus (FEEC) and operator learning, focusing on the crucial aspect of physical structure preservation.

- Clear Identification of a Problem: The paper successfully draws attention to a potential pitfall in the CWF framework (the possibility of fulfilling conservation laws in a physically meaningless way).

- Practical Solution: The proposed flux regularization is a simple and intuitive first step to address the identified problem.

**Weaknesses:**

**Major Technical Weakness**

- Limited Conceptual Contribution: The core observation, that the original CWF constraint is underdetermined and allows trivial solutions, is more of a theoretical oversight in the original work than a novel insight. The proposed fix (adding a flux reconstruction loss) is a straightforward application of multi-task learning and does not introduce a new methodological or theoretical framework.

- Superficial Theoretical Contribution: The core theoretical contribution (Propositions 1 and 2), while mathematically correct, is conceptually shallow. It demonstrates that an *unconstrained* flux field $\hat{f}$ can be computed in a post-hoc manner to satisfy a discrete conservation law for any given $u$ and $s$. However, in the original CWF formulation by Kinch et al., the flux is \emph{not} unconstrained; rather, it is parametrized by a neural network $\mathcal{NN}(\hat{u}; \phi, z_i)$, whose structure and training are tightly coupled with the learning of the partition of unity. The paper fails to prove that this *coupled, learned* system will inevitably converge to the trivial solutions described in the propositions. It only shows that such solutions *exist*, which is a significantly weaker claim. As a result, the presented ``theoretical insight'' is better characterized as an \emph{observation} regarding underdetermined linear systems, rather than a rigorous analysis of the CWF learning dynamics. Although two propositions are included on the underdetermined nature of the conservation constraint, these results are elementary from a linear algebra perspective and do not offer new theoretical understanding of the approximation properties or behavior of CWFs.

**Major Conceptual Weakness**

- Solution Lacks Physical Intuition and Rigor: The proposed solution, adding an L2-loss on the flux, reduces a hard physical constraint (the conservation law) to a soft, data-fitting penalty. This undermines the primary motivation of using CWF, which is to provide **guaranteed structure preservation**. The method now relies on the balance of a hyperparameter $\(\lambda\)$ to *approximately* satisfy physics, a common paradigm in Physics-Informed Neural Networks (PINNs) which the authors implicitly critique. A more principled approach would be to impose structure on the flux network $\(\mathcal{NN}\)$ itself (e.g., enforcing symmetry or other physical properties) to restrict the solution space without sacrificing the hard constraint.

- Insufficient Analysis of Trade-offs: The paper does not adequately address the trade-off between structure preservation and model expressivity. For instance, the flux regularization term may force the model to learn overly smooth or simplistic flux representations, especially in highly nonlinear regimes, a concern only briefly mentioned in Appendix C.


**Major Technical Flaw**

 - Inconsistent Error Metrics Suggest Fundamental Issues: The reported error metrics in Table 1 reveal a fundamental inconsistency that undermines the technical validity of the results. The distribution errors (e-7 to e-10) and flux errors (e-1 to e+0) differ by 7-10 orders of magnitude, which is mathematically implausible given the physical relationship between these quantities ($f$ contains $\nabla u$). Until this discrepancy is rigorously explained and validated, the central claim, that the original CWF formulation leads to trivial conservation, rests on unreliable evidence, as the observed effect could be an artifact of the evaluation method or numerical instability rather than a profound physical insight. The authors must demonstrate that their error metrics on their $u$ and $f$ predictions are mathematically consistent and that the massive flux error is not a mere consequence of a poorly conditioned numerical scheme.

- Lack of Meaningful Baselines: The comparison is limited to the regularized vs. unregularized versions of their own model. The work lacks comparisons against strong and relevant baselines, such as a pure data-driven model (e.g., a standard Transformer predicting both $\(u\)$ and $\(f\)$) or other physics-constrained architectures (e.g., conservative PINNs, finite volume-inspired methods). Without this, it is impossible to gauge the actual benefit of the complex CWF machinery over simpler approaches.

- Toy Problems and Linearity: The experiments are conducted predominantly on linear or weakly nonlinear advection-diffusion problems. The claim of enabling "physics recovery" is not sufficiently tested against highly nonlinear or chaotic systems, where the identified problem and proposed solution would face a much greater challenge. There is no evidence that the method scales to 3D, complex geometries, or more challenging PDE systems (e.g., Navier-Stokes, wave equations). This severely limits the claimed generality of the approach.

- Grid Dependency: The method is fundamentally tied to the fixed, underlying "fine-grained partition of unity." The paper does not address or even acknowledge the limitation this imposes. It is unclear how the approach would generalize to synthetic datasets and real-world scenarios where training and test data come from different meshes or geometries, a key selling point of neural operators.


**Minor issue**

- Inadequate Discussion of Related Work: The discussion of structure-preserving methods is superficial. It does not engage deeply with the trade-offs between different approaches. For instance, how does the performance and guarantee of this method compare to methods that directly discretize and solve the PDEs in the loss function (e.g., FEM-based variational losses)? The claim that CWFs are the *only* framework that allows this transformation with "minimal architectural intervention" is strong but unsupported by a thorough comparison.

**Questions:**

- The propositions assume an unconstrained $\(\hat{f}\)$. However, in practice, \(\hat{f}\) is output by a neural network conditioned on $\(\hat{u}\)$. Can you provide empirical or theoretical evidence that the *coupled learning process* of $\(W(z;\theta)\)$ and $\(\mathcal{NN}(\hat{u}; \phi, z_i)\)$ is inherently biased towards the "trivial" solutions you construct, rather than just being capable of representing them?

- By relaxing the hard physics constraint to a soft penalty, the method no longer provides exact conservation. How do you reconcile this with the stated goal of "structure preservation"? What are the quantitative trade-offs between the flux reconstruction accuracy and the exact satisfaction of the conservation law?

- How does the method compare to other physics-informed or structure-preserving approaches (e.g., conservative PINNs, finite volume networks) in terms of accuracy, stability, and computational cost? A comparison with pure data-driven models should also be included.

- The method is intrinsically dependent on a pre-defined, fixed mesh for the fine-grained partition. How would you apply this approach to a synthetic/real-world dataset where inputs come from irregular geometries or inconsistent discretizations? Does this not severely limit the applicability of the method as a general neural operator?

- Have the author tested the method on more complex systems (e.g., 3D flows, systems with shocks, or multiphysics problems)? If not, what are the anticipated challenges?

- Given that the flux regularization is a simple L2-loss, what is the novel *architectural* or *theoretical* insight here beyond a standard multi-task learning setup? Why is the CWF framework necessary to implement this idea?

- The paper included one sensitivity study of the choice of $\lambda$, and is there a principled way to select it for a new problem?

- Explain why the unregularized CWF achieves near-perfect distribution reconstruction while failing completely at flux reconstruction. The significant discrepancy in the magnitude of errors between the two results raises concerns about their reliability.

- Present the main architecture diagram of the model. Provide information on computational efficiency (e.g., the running time and memory) and the detailed model configuration summary table.

---

### Official Review · Reviewer_pg3J · 2025-11-10

**Soundness:** 2
**Presentation:** 1
**Contribution:** 2
**Rating:** 2
**Confidence:** 3

**Summary:**

The authors look at conditional Whitney forms. This is an approach inspired by finite element analysis, and combined with machine learning in the context of recovering PDE physics and reduced-order modeling. The authors look at this framework within the context of advection-diffusion models.

**Strengths:**

- The text in the paper is mostly well-written.

- The problem area and approach, including in the context of reduced-order modeling, seems under explored and could be promising.

**Weaknesses:**

- The paper is hard to follow in terms of understanding the exact motivations and significance of the problem being solved. Are all the problems in an operator learning setting of mapping parameters to solutions? What advantage is ML giving here that wasn’t possible before? What is this in terms of speed and accuracy?


- It’s hard to tell the exact motivations of the problem setting and the ML approach taken. Why do the authors pick the ML training and problem setting that they choose? What worked, and didn’t work in these problem settings? Ablations would be helpful here.

- The way the authors are writing this paper, they seem to be assuming that the reader is familiar already with the idea of conditional Whitney forms. But, the paper that introduces and discusses this is a paper that was only posted on arxiv in August ’25 (this year). This paper should discuss this idea in a standalone format, especially as the paper being referred to is a very recent one.

- The examples shown here seem fairly toy examples. Would the utility of such an approach, in the reduced-order modeling context, be more relevant for 3D problems?

**Questions:**

- Can you provide a lot more detail on the problem motivation?

- Can you provide a lot more detail giving more insight into the general motivations of the ML approach, including ablations?

- How does this method position itself in terms of other methods combining insights from finite element methods and ML?

- How do results change when there is more data?

---

### Note · Authors · 2025-11-14

I have read and agree with the venue's withdrawal policy on behalf of myself and my co-authors.